# Quantification of Visceral Fat at the L5 Vertebral Body Level in Patients with Crohn’s Disease Using T2-Weighted MRI

**DOI:** 10.3390/bioengineering11060528

**Published:** 2024-05-22

**Authors:** Favour Garuba, Aravinda Ganapathy, Spencer McKinley, Karan H. Jani, Adriene Lovato, Satish E. Viswanath, Scott McHenry, Parakkal Deepak, David H. Ballard

**Affiliations:** 1School of Medical Education, Washington University School of Medicine in St. Louis, St. Louis, MO 63110, USA; f.garuba@wustl.edu (F.G.); aganapathy@wustl.edu (A.G.);; 2Mallinckrodt Institute of Radiology, Washington University School of Medicine in St. Louis, St. Louis, MO 63110, USA; jani@wustl.edu (K.H.J.); lovato@wustl.edu (A.L.); 3Department of Biomedical Engineering, School of Engineering, Case Western Reserve University, Cleveland, OH 44106, USA; satish.viswanath@case.edu; 4Division of Gastroenterology, Washington University School of Medicine in St. Louis, St. Louis, MO 63110, USA; smchenry@wustl.edu (S.M.); deepak.parakkal@wustl.edu (P.D.)

**Keywords:** MRI, visceral fat, L5, umbilicus, Crohn’s

## Abstract

The umbilical or L3 vertebral body level is often used for body fat quantification using computed tomography. To explore the feasibility of using clinically acquired pelvic magnetic resonance imaging (MRI) for visceral fat measurement, we examined the correlation of visceral fat parameters at the umbilical and L5 vertebral body levels. We retrospectively analyzed T2-weighted half-Fourier acquisition single-shot turbo spin echo (HASTE) MR axial images from Crohn’s disease patients who underwent MRI enterography of the abdomen and pelvis over a three-year period. We determined the area/volume of subcutaneous and visceral fat from the umbilical and L5 levels and calculated the visceral fat ratio (VFR = visceral fat/subcutaneous fat) and visceral fat index (VFI = visceral fat/total fat). Statistical analyses involved correlation analysis between both levels, inter-rater analysis between two investigators, and inter-platform analysis between two image-analysis platforms. Correlational analysis of 32 patients yielded significant associations for VFI (r = 0.85; *p* < 0.0001) and VFR (r = 0.74; *p* < 0.0001). Intraclass coefficients for VFI and VFR were 0.846 and 0.875 (good agreement) between investigators and 0.831 and 0.728 (good and moderate agreement) between platforms. Our study suggests that the L5 level on clinically acquired pelvic MRIs may serve as a reference point for visceral fat quantification.

## 1. Introduction

Modern obesity research has progressed beyond studying the impact of body mass index and fat mass on health and disease outcomes to analyzing the effects of fat distribution, which is a better prognostic biomarker [1]. This has led to the discovery that a higher proportion of visceral fat is associated with poorer outcomes in various disease states, including cardiovascular disease, diabetes, and inflammatory bowel disease [2,3,4], due to the production of higher levels of pro-inflammatory cytokines and adipokines compared to subcutaneous fat [5]. Consequently, research involving the quantification of visceral fat to determine clinical outcomes has expanded in the last decade and techniques to more accurately and reliably quantify visceral fat are in high demand.

These techniques involve imaging-based fat segmentation with computed tomography (CT) or magnetic resonance imaging (MRI), which have successfully been utilized to quantify visceral and subcutaneous fat in prior studies with similar levels of accuracy [6,7]. The widespread use of these imaging modalities in evaluating common acute and chronic pathologies also makes them easily accessible methods of determining fat distribution. Furthermore, when using semi-automated and fully automated fat segmentation software, the fat quantification process can be completed in a relatively short amount of time [8,9]. 

MRI is less frequently utilized due to a less defined intensity value range specific to adipose tissue compared to CT, especially when utilizing T2-weighted sequences. It is also associated with other challenges, such as limited abdominal and pelvic MR matrix sizes that do not include peripheral superficial fat in the field of view [10]. Nevertheless, MRI may be the preferred modality when aiming to limit ionizing radiation, leading to the use of T1 fat-only images acquired and reconstructed using the Dixon technique [11,12]. However, due to the absence of Dixon sequences in certain abdominal and pelvic MR protocols, some studies have explored T2-weighted half-Fourier acquisition single-shot turbo spin echo (T2 HASTE) as a potential alternative [13,14].

Regardless of the imaging modality utilized, segmenting at the third lumbar level (L3) or the umbilicus is the widely accepted method to analyze fat distribution for outcomes research when utilizing single-slice measurements [15]. However, there are certain patient populations that are more likely to obtain pelvic images where both levels are unavailable for fat segmentation. For example, perianal Crohn’s disease patients tend to undergo pelvic MRIs to assess for perianal complications like perianal fistulas [16]. In such patient populations, segmenting at the level of the fifth lumbar vertebra (L5) may be the only option. As a result, investigating alternative landmarks, such as the level of L5, becomes necessary to establish a reliable and consistent approach to quantify adipose tissue distribution in this specific patient subgroup.

Therefore, this study aims to explore the correlation between visceral adipose tissue parameters obtained at the level of the umbilicus and L5. A secondary objective was to assess the feasibility of quantifying fat deposits utilizing T2 HASTE MRI.

## 2. Materials and Methods

### 2.1. Patient Cohort 

We conducted a single-center retrospective study in accordance with the Health Insurance Portability and Accountability Act, as approved by our institutional review board (IRB). The informed consent requirement was waived due to the study’s retrospective design (local IRB approval numbers 202204095 and 202307200). Patients with stricturing Crohn’s disease who underwent an MRI enterography examination of the abdomen without and with contrast from 2018 to 2020 were selected from a larger dataset of prospectively enrolled patients with Crohn’s disease starting various advanced therapies. Exclusion criteria were: (i) patients without T2-weighted HASTE sequence images, (ii) patients with an MRI field of view that was not large enough to capture their entire body habitus, leading to ‘out-of-view’ fat data that could not be estimated, and (iii) patients with poor-quality imaging due to excessive artifact. Demographic information such as age, sex, weight, height, and BMI were obtained from MRI data stored on our institution’s PACS interface (Sectra IDS7 version 25.1, Linköping, Sweden). For patients with missing data on the PACS interface, chart review using our institution’s electronic health record system (EPIC Hyperspace) was utilized to obtain the data from a clinical note closest to the date of the MRI. 

### 2.2. Adipose Tissue Quantification 

Two study investigators (FG and AG; medical students) were trained and supervised by a board-certified abdominal radiologist (DHB; an abdominal radiologist with three years post-fellowship experience) to quantify the amount of subcutaneous fat and visceral fat at a single slice. The single-slice measurements were obtained at the level of the umbilicus and L5. T2-weighted HASTE MRI axial sequences without fat saturation were utilized for all patients. MRI was performed on clinical Siemens scanners, including 1.5T (Espree, Aera, Sola) and 3T (Skyra, Vida) systems. Axial HASTE MRI sequences were acquired with the following parameters: (i) a resolution of 320 × 320 pixels, (ii) a slice thickness (ST) of 4 mm with a 1 mm inter-slice gap, and (iii) a repetition time/echo time (TR/TE) range of 1000–1300 milliseconds/90 milliseconds. In this study, subcutaneous fat was defined as adipose tissue within the subcutaneous space that was superficial to the abdominal and back muscles, while visceral fat was defined as adipose tissue within the abdominal cavity excluding intramuscular fat in the abdominal and back muscles. Fat segmentation was completed on two freeware platforms to assess consistency across platforms. Images uploaded to both interfaces were exported from our institution’s PACS interface as a Digital Imaging and Communications in Medicine (DICOM) image using the anonymization protocol, which removes all personal health information from the images. To obtain the visceral fat parameters and perform an inter-rater and inter-platform analysis, MRIs of all included patients were segmented on platform 1 by one study investigator and MRIs of a subset of randomly selected patients were segmented on platform 2 by two investigators. Platform 1 is known as CoreSlicer [17] and can be accessed via https://coreslicer.com/ (accessed on 21 February 2024). It measures subcutaneous fat area and visceral fat area using an artificial intelligence model that performs a region of interest (ROI) analysis based on intensity range. Platform 2 is known as Parametric Magnetic Resonance Imaging v1.2.31-b (pMRI) and can be downloaded via the software’s website at https://www.parametricmri.com/ (accessed on 22 February 2024). It utilizes a volumetric analysis that measures subcutaneous fat volume and visceral fat volume from a single slice by factoring in slice thickness. Platform 1 reports its measurements in cm^2^ while platform 2 reports its measurements in cm^3^. For data obtained from both platforms, the visceral fat index (VFI = amount of visceral fat/amount of visceral and subcutaneous fat) and visceral fat ratio (VFR = amount of visceral fat/amount of subcutaneous fat) were calculated. The utilization of relative adipose tissue parameters, such as VFI and VFR, permits standardization of results obtained from both platforms due to its focus on patient-specific proportions of visceral fat rather than absolute values. 

### 2.3. Platform 1 Image Analysis

Images of all patients were segmented at the level of the umbilicus and L5 by one study investigator (FG; medical student). After uploading the DICOM image to the CoreSlicer website, the level of segmentation was selected. Subsequently, the intensity range for subcutaneous adipose tissue was selected. The intensity range varied per patient as it was selected based on visual analysis. A brush tool was then utilized to select the region of interest (ROI). The ROI for both subcutaneous fat and visceral fat was selected according to the study’s definition of both deposits. During this step, the intensity ranges were adjusted to maximize inclusion of adipose tissue and exclusion of non-adipose tissue. Intensity range selection and ROI designation were repeated for visceral adipose tissue (Figure 1). The subcutaneous fat area (SFA) and visceral fat area (VFA) were recorded. The total fat area (TFA = SFA + VFA), VFR (VFR = VFA/SFA), and VFI (VFI = VFA/TFA) were subsequently calculated. The data obtained from this platform were utilized for the correlation analysis. 

### 2.4. Platform 2 Image Analysis

After a 2 week wash out period, the images were segmented on platform 2 by both medical student investigators [FG and AG] for the inter-platform and inter-rater analysis. The images were loaded into pMRI and segmented at the level of the umbilicus and L5 using the volumetric analysis and segmentation module. Using a region of interest (ROI) selection tool, two ROIs were selected for subcutaneous fat and visceral fat. The signal intensity thresholds were manually set to quantify the amount of adipose tissue within the ROIs. The signal intensity-based segmentation was then reviewed and revised to correct for misclassified tissues (Figure 1). Both study investigators completed this segmentation process for a smaller cohort of randomly selected patients (*N* = 15) from the original cohort. The slice closest to the center of each level was utilized by both investigators. After each segmentation, subcutaneous fat volume (SFV) and visceral fat volume (VFV) were recorded for each slice. Total fat volume (TFV = SFV + VFV), VFR (VFR = VFV/SFV), and VFI (VFI = VFV/TFV) were subsequently calculated. 

The steps of segmentation on platform 1 and 2 are displayed in Figure 1. Each segmentation result was reviewed by the supervising study radiologist and corrected as needed.

### 2.5. Estimation of Field of View Restriction 

After segmentations, the field of view restrictions were estimated for images that had portions of their subcutaneous fat out of screen view. The estimates were completed by calculating the linear distance from the cutoff to the body wall was using a coronal image that included all of the body habitus. The localizer image was utilized as the preferred means of measuring the linear distance. If the estimation could not be completed using the localizer image, any other coronal image that included all of the patient’s habitus was utilized. These measurements were completed on our institution’s PACS interface, which enables synchronization of multiple images. The image slice used for segmentation was synced with the selected coronal image to enable measurement of the linear distance of subcutaneous fat from the body wall on both the axial and the coronal slice. The difference between the linear distances on both images was subsequently calculated and utilized to estimate the amount of ‘out-of-view’ fat data on the axial slice that was segmented. These linear distances were solely utilized to objectively estimate the amount of missing fat data. No further calculations were executed to attempt to extrapolate the total volume of subcutaneous fat. A sample is outlined in Figure 2.

### 2.6. Statistical Analysis

Linear regression analysis was used to investigate the correlations between the VFI and VFR at two anatomical levels: L5 and umbilicus. Data were fitted using the least squares method to determine the best fitting line, which was then plotted with 95% confidence interval bands. If any patient was excluded from the main analysis due to field of view restriction, a regression analysis was also conducted with the inclusion of these patients to assess if inclusion of these patients would affect the study results. Inter-rater and inter-platform reliability was assessed by two-way intraclass coefficient (ICC) analysis and an F test was conducted for significance testing. ICC values were also calculated between T2 HASTE and Dixon values to validate T2 HASTE as an alternative sequence. Normality was assessed using the Shapiro–Wilk test. T-tests were used for pairwise comparisons when data were normal, and Wilcoxon rank-sum test was used when data were non-normal. Data analysis was conducted in Rstudio version 2023.06.1 using package cufunctions [18] and Graphpad Prism version 10.0.0 for Windows, GraphPad Software, Boston, MA, USA, www.graphpad.com (accessed on 21 January 2024). Figures were created using Graphpad Prism. 

## 3. Results

### 3.1. Study Population Characteristics

A total of 41 patients with Crohn’s disease were evaluated for inclusion in this study. Of these, nine patients were excluded as summarized in Figure 3. For patients with a field of view restriction that we were able to estimate, the linear distance from the image cutoff to their body wall ranged from 0.19 cm–4.07 cm. The only patient with an estimate greater than the aforementioned range was excluded from further analysis as an outlier. The final cohort consisted of 32 patients (18 female; 14 male). Baseline characteristics of all patients are portrayed in Table 1.

### 3.2. Subcutaneous and Visceral Fat

Mean SFA at the umbilical level was 239 ± 96.3 cm^2^ while mean SFA at L5 was 262 ± 97.3 cm^2^ (*p* = 0.349). Mean VFA at the umbilical level was 106 ± 68.2 cm^2^ while mean VFA at L5 was 95.4 ± 45.8 cm^2^ (*p* = 0.8932). Mean VFI at the umbilical level was 0.3 ± 0.105 while mean VFI at L5 was 0.273 ± 0.0881 (*p* = 0.256). Mean VFR at the umbilical level was 0.465 ± 0.256 while mean VFR at L5 was 0.395 ± 0.177 (*p* = 0.2767). Fat parameters obtained from platform 1 analysis of all study patients are demonstrated in Table 2. A case example is portrayed in Figure 4. 

### 3.3. Correlation Analysis of Fat Measurements between Umbilicus and L5

Correlational analysis between L5 and umbilicus levels yielded significant associations for both VFI (r = 0.85; *p* < 0.0001) and VFR (r = 0.74; *p* < 0.0001). These correlational findings are depicted in Figure 5 and Figure 6. 

Correlational analysis was also conducted using the entire patient cohort, without exclusion of the six patients with field of view restrictions that we were unable to estimate and the single patient whose field of view restriction was an outlier compared to that of patients included in the main analysis. As prior, VFI (r = 0.84; *p* < 0.0001) and VFR (r = 0.74; *p* < 0.0001) both demonstrated significant associations. These findings are depicted in Appendix A.

### 3.4. Inter-Platform and Inter-Rater Analysis

The inter-rater ICC values for VFI and VFR between the two investigators were 0.846 and 0.875, respectively, which depict good reliability (0.75–0.9) [19]. To visualize the variation in fat segmentation measurements between the two raters, the data collected are plotted against each other in Figure 7. All fat parameters from the inter-rater analysis are outlined in Table 3, which also shows that the difference between the fat parameters obtained from the level of the umbilicus and those obtained from the level of L5 is non-significant. The average time spent on segmentation by the raters was 12.09 ± 2.75 min for the initial analyst (FG) and 13.3 ± 3.15 min for the inter-observer (AG).

Inter-platform agreement analysis was also conducted to assess the variation in VFI and VFR measurements between the two platforms used for segmentation measurements. The ICC values for the VFI and VFR between the two platforms were 0.831 and 0.728, respectively, which depict good and moderate reliability [19]. The data collected are plotted against each other in Figure 8. 

Only four patients had Dixon and T2 HASTE sequence images at the umbilical and L5 levels. Two-way ICC analysis yielded a value of 0.914 for VFR, and a value of 0.962 for VFI, depicting excellent reliability. 

## 4. Discussion

The umbilicus or L3 is most often utilized as a reference point for fat quantification when using CT or MRI axial images. These levels may be inaccessible in certain patient cohorts that mainly undergo pelvic imaging, like patients with perianal Crohn’s disease. We aimed to demonstrate the feasibility of using pelvic MRIs for fat quantification by assessing the correlation between relative visceral fat parameters obtained at the umbilical and L5 levels and discovered that VFR and VFI from the L5 level strongly correlates with the values obtained from the umbilical level. 

Prior research points to a strong correlation between adipose fat distribution and metabolic syndrome, cardiovascular disease, and vascular dysfunction [2,3,5]. Thus, accurate quantification of visceral adiposity is valuable in assessing an individual’s susceptibility to these health concerns. Often, when quantifying visceral adiposity, L3 or the umbilicus has conventionally served as the standard level for segmentation in evaluating visceral fat [15,20]. However, one prior study has demonstrated the potential of L5 for this purpose [21]. In some patient groups, like patients with perianal Crohn’s disease who undergo fistula protocol pelvic MRI [16], this may be the only feasible level that is available, thus it is important to explore how visceral fat parameters obtained from the level of L5 correlate with those obtained from the level of the umbilicus or L3. The umbilicus was utilized as the standard reference point in this study due to ease of identification as a visual landmark that does not require the extra step of counting vertebral levels. Although L3 is considered a less variable reference compared to the umbilicus, whose location may change based on surgical history, visceral fat measurements between both levels are very strongly correlated [22]. Also, while previous studies have compared absolute visceral fat area or volume obtained at the levels of L2–L5 [15,20,23], our study compares relative visceral fat parameters, such as VFR and VFI, at the level of the umbilicus and L5. These ratios may permit standardization of visceral fat parameters obtained from different platforms and allows research investigators to compare relative proportions of visceral fat between various individuals by providing an internal control for total adiposity for each patient. 

Our analysis on the correlational relationship of VFI and VFR between the umbilical and L5 levels provides valuable insights into the potential utilization of pelvic MRIs to quantify visceral fat in specific patient cohorts. Although multiple studies have demonstrated a correlation between single slice visceral fat area at the umbilicus and total visceral abdominal fat volume [24,25,26], our results prove that L5 can be a suitable alternative anatomical landmark in the absence of access to the umbilicus. The strong correlation of VFI (r = 0.85) and VFR (r = 0.74) between the umbilical and L5 levels in our study suggests that measurements taken at the level of L5 can serve as a potential proxy for those typically taken at the umbilicus level. These results are also in line with the findings from prior articles that utilized CT images for fat quantification. One study demonstrated high correlations of adipose tissue parameters at the third, fourth, and fifth lumbar level, although the level of the umbilicus was not considered in that study [27]. Another study also demonstrated that measures of visceral adipose tissue at L2–L3, L3–L4, and L4–L5 possess similar associations to metabolic risk, with odds ratios of 2.7, 2.8, and 2.6, respectively [21]. Therefore, although the level of L3 or the umbilicus is the widely accepted level for single-slice fat segmentation, other lumbar vertebral levels, such as L5, have shown strong correlations with metabolic risk and may be utilized as an alternative when unable to complete a whole-body fat volumetric assessment. 

With regard to our secondary purpose of assessing the feasibility of visceral fat quantification using T2 HASTE MRI axial sequences without fat saturation, the strong inter-rater and inter-platform reliability establishes the consistency and dependability of visceral fat measurements obtained using T2 HASTE axial images. Most studies utilizing MRI for fat quantification depend on T1-weighted fat-only Dixon images; however, this is one of the few studies that utilized T2-weighted MRI for fat quantification. One study that quantified visceral adipose tissue at the level of the umbilicus using T1-weighted and T2-weighted MRI showed that T2-weighted images have slightly higher inter-observer reproducibility compared to T1-weighted images [28]. In another study that derived subcutaneous and visceral adipose tissue reference values for children using T2-weighted images, the inter-observer agreement between randomly selected sets of MRI was also high, with a mean difference of 0.07 cm^2^ and 0.08 cm^2^ for subcutaneous adipose tissue and visceral adipose, respectively [13]. Using T2 HASTE is also advantageous due to its standard inclusion in most MRI protocols and its reduced motion artifact compared to the Dixon technique. The resultant increase in data quality and availability make T2 HASTE a strong candidate sequence in any research that uses MRI for fat quantification, especially when uniform access to fat-only Dixon is limited. Its excellent reliability based on inter-sequence ICCs of 0.914 and 0.962 for VFR and VFI, respectively, also supports its strong potential for use in fat quantification studies as an alternative to fat-only Dixon. Therefore, the utilization of T2 HASTE sequence can be considered when executing comprehensive visceral fat quantification research with limited access to fat-only Dixon MRI.

It is important to also discuss the challenges of using MRI in fat segmentation as opposed to CT. First, CT imaging is more standardized than MRI in respect to fat segmentation due to the relatively consistent fat-specific Hounsfield unit range, in contrast to MRI signal intensities which have no defined ranges for adipose tissue, especially when using T2-weighted sequences without fat saturation. The range of values representing adipose tissue varies between scanners, depends on pulse sequences, and is influenced by the hardware used, leading to the spatially varying signal intensity non-uniformity that makes standardization more challenging [10]. These challenges can lead to misclassification of fat and other tissues during the segmentation process, which emphasizes the importance of employing semi-automatic or automatic programs that can help clinicians or researchers limit such mistakes. For example, the pMRI volumetric analysis module permits analysts to change the opacity of volumetric overlays, which allows for ease of correction of misclassified tissues after selecting intensity thresholds. Other available tools for semi-automatic or automatic fat quantification using MRI include deep learning networks, such as RAdipoSeg and FatSegNet, and commercial medical image analysis software, such as sliceOmatic [7,11,29]. MRI is also more prone to issues with artifacts due to restriction of edge of field of view due to matrix sizes that do not include peripheral superficial fat in the field of view. Further, it has been shown that single MRI slices can struggle with accurately predicting changes in visceral and subcutaneous adipose tissue changes, such as in the setting of weight loss [30]. It is for these reasons that MRI is utilized less frequently than CT in fat quantification studies. However, MRI may be the most feasible modality for certain patient populations, especially when aiming to limit ionizing radiation. Potential solutions to improve the quantification of adipose tissue with MRI include the use of complementary data such as T1 or T2 mapping and proton density fat fraction rather than solely depending on raw signal intensity [31,32]. 

In regard to the limitation of MR acquisition matrix sizes, which may exclude portions of the subcutaneous adipose tissue from the field of view, the patients that are most affected are those with high degrees of obesity, leading to ‘out-of-view’ fat data [33]. The solution to this issue is not as simple as excluding all patients with such field of view restrictions as that may affect the generalizability of results to patients with a larger body habitus. Solutions that have been proposed to fully quantify subcutaneous fat in these patients include the use of surrogate measures, such as simple lengths of subcutaneous tissue from the hip girdle to a non-restricted body wall [33]. Regardless, results from our supplementary analysis show that inclusion of patients with large exclusions of their body habitus (patients with greater than a 6 cm linear distance exclusion from their body wall and patients with exclusions that we were unable to estimate due to larger body habitus) would not affect the significance of this study’s results. Further studies are needed to assess how these restrictions of field of view affect the determination of metabolic risk and clinical outcomes and how to mitigate these restrictions in mixed cohorts where some patients have these restrictions and some patients do not.

The main limitation of our study method is its retrospective study design based on patients with Crohn’s disease at a single center. This may affect the generalizability of our study to healthy individuals or other patient cohorts. Our small cohort size also contributes to lower statistical power, which may make it difficult to determine true estimates of correlation. We are also aware that an optimal study design would typically involve visceral fat quantification with full-body Dixon MRI in combination with full-body CT to study total body fat; however, this was infeasible with clinically routine sequences and a retrospective study design. For the inter-platform analysis, we utilized two distinct freeware platforms with likely differing algorithms for the computation of visceral fat measurements, which may explain the relatively lower inter-platform ICCs for VFI and VFR. However, we were unable to decipher which specific differences between the algorithms contributed to the relatively lower ICCs. For future larger studies, automatic artificial intelligence-based algorithms are recommended. For the direct comparison between visceral fat measurements from Dixon and T2 HASTE sequences, our sample size was too small for meaningful statistical correlation analyses. Such comparison will also benefit from further investigation via a larger prospective study to corroborate our findings. For the inter-platform and inter-rater subset analysis, focusing on a representative subset for segmentation restricts the depth of variability analysis, which may limit the broader applicability. Finally, the focus of this study was to analyze the correlation between visceral fat parameters between the umbilical and L5 levels; we did not assess how our results affect true metabolic risk or outcomes in our patient cohort.

## 5. Conclusions

L5 can serve as an alternative anatomical landmark to the umbilicus when quantifying visceral fat using clinically acquired pelvic MRIs. In addition, quantifying visceral fat using T2 HASTE MRI axial sequence without fat saturation is feasible.

## Figures and Tables

**Figure 1 bioengineering-11-00528-f001:**
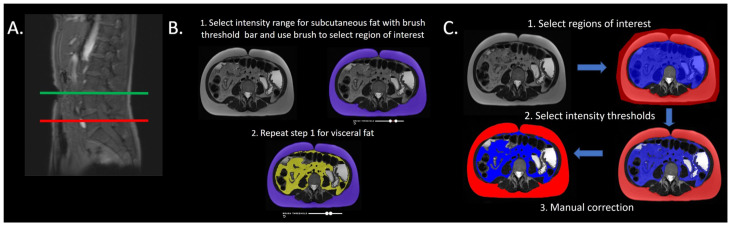
The process of semi-automatic fat segmentation using CoreSlicer (Platform 1) and pMRI (Platform 2). (**A**) Each patient’s MRI was segmented at the level of the umbilicus (green line) and the level of L5 (red line). (**B**) The steps of subcutaneous fat (outlined in purple) quantification and visceral fat (outlined in yellow) quantification on platform 1. (**C**) The steps of subcutaneous fat (outlined in red) quantification and visceral fat (outlined in blue) quantification on platform 2.

**Figure 2 bioengineering-11-00528-f002:**
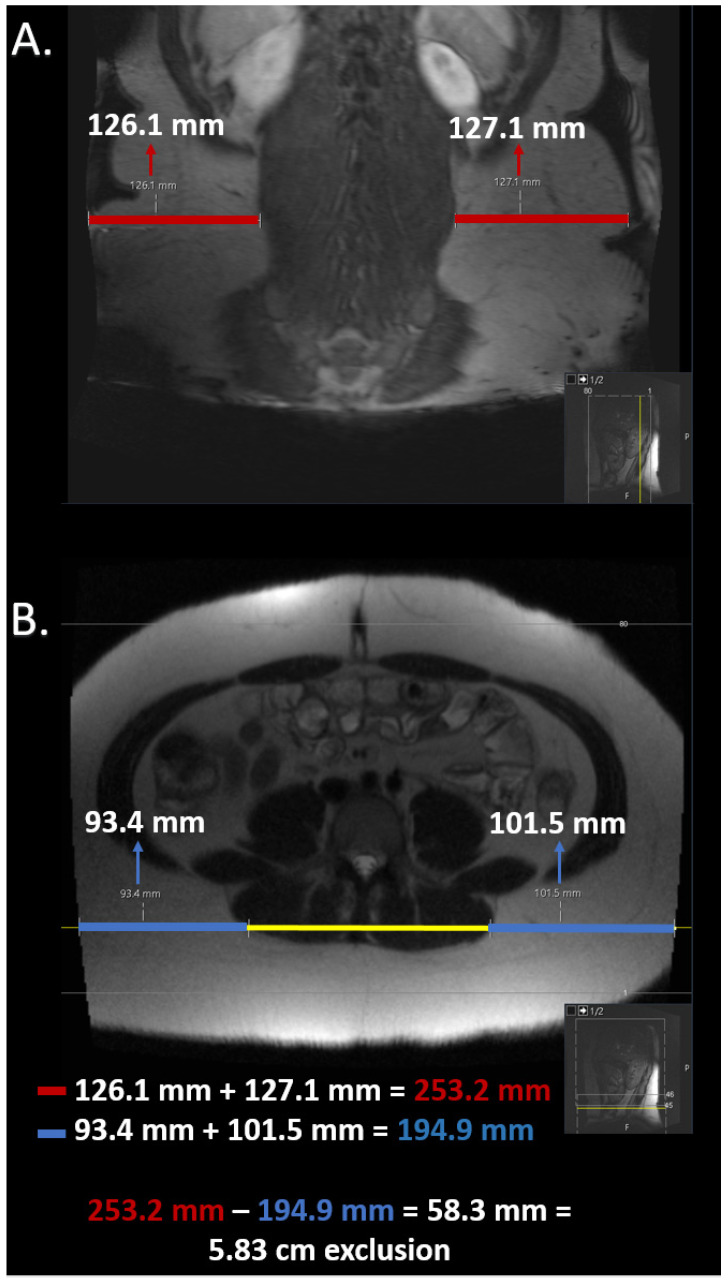
Estimation of field of view restriction in a patient with a large portion of their MR axial image out of screen view. (**A**) portrays the coronal localizer image while (**B**) portrays the T2-weighted HASTE sequence without fat saturation axial image that was segmented. The red lines in (**A**) highlight the linear distance of subcutaneous fat from the body wall in the coronal localizer image. The blue lines in (**B**) highlight the linear distance of subcutaneous tissue from the cutoff point in the axial image. The yellow line in (**B**) highlights the line that appears after syncing the axial and coronal slice to confirm where the coronal slice is located in the axial slice. The linear distance of subcutaneous fat in B was subtracted from the linear distance of subcutaneous fat in A to provide an estimate of the linear distance of the subcutaneous fat that was cut off from the axial image.

**Figure 3 bioengineering-11-00528-f003:**
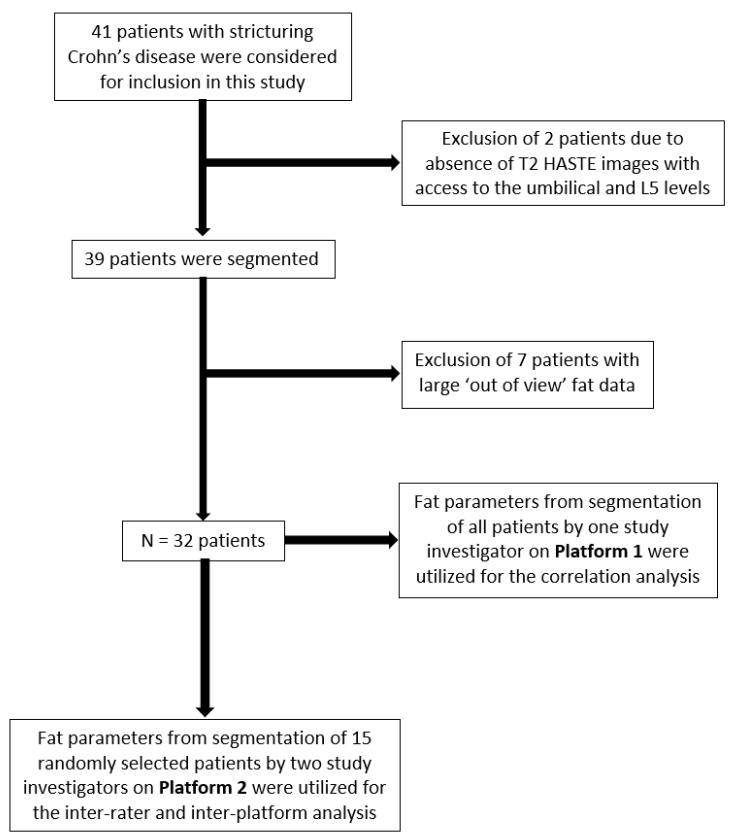
Study flow chart.

**Figure 4 bioengineering-11-00528-f004:**
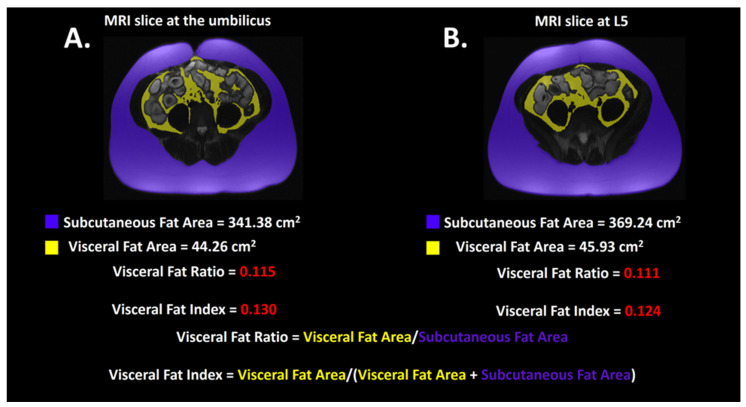
A case example of fat segmentations at the level of the umbilicus in (**A**) and the level of L5 in (**B**). On these T2-weighted HASTE sequence without fat saturation axial images, subcutaneous adipose tissue is highlighted in purple while visceral adipose tissue is highlighted in yellow.

**Figure 5 bioengineering-11-00528-f005:**
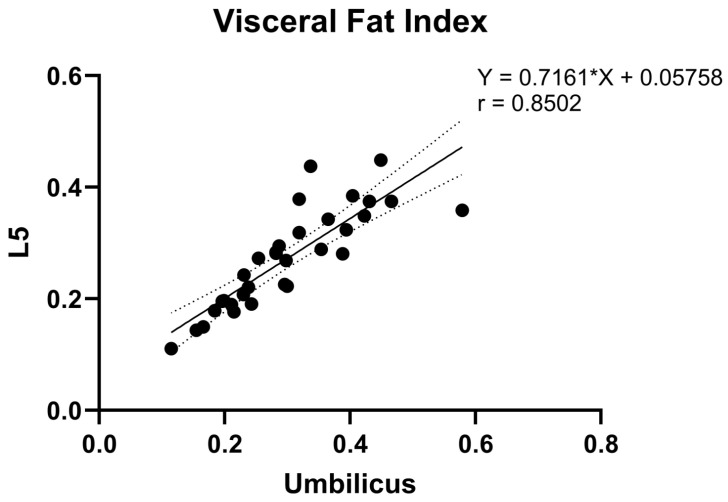
Correlation of visceral fat index measurements at L5 vs. umbilicus levels.

**Figure 6 bioengineering-11-00528-f006:**
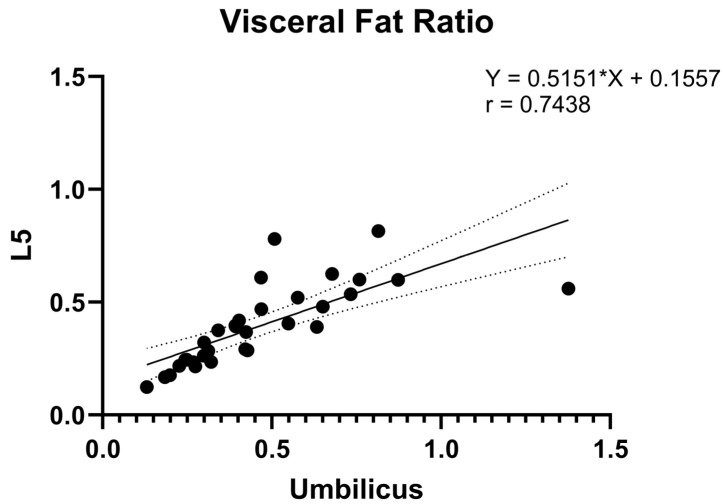
Correlation of visceral fat ratio measurements at L5 vs. umbilicus levels.

**Figure 7 bioengineering-11-00528-f007:**
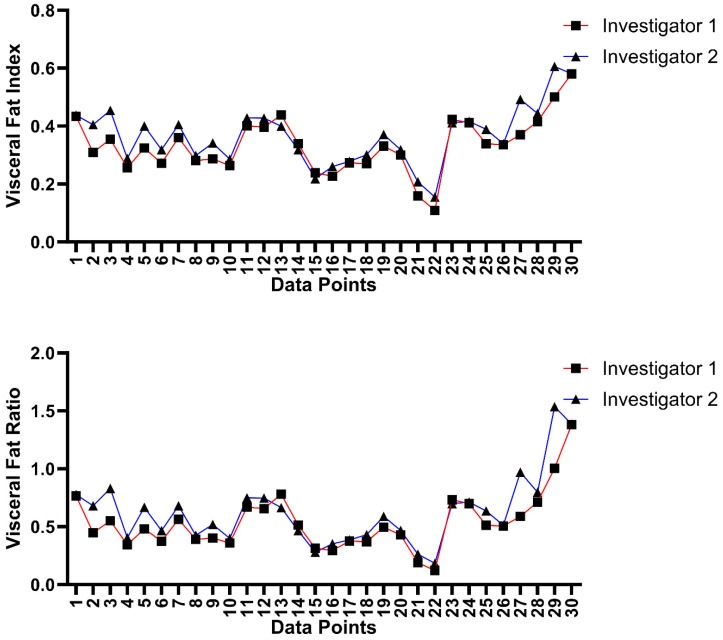
Inter-rater variation in visceral fat ratio and visceral fat index measurements. “Data Points” refers to scans taken at L5 and umbilicus levels for each of the 15 patients.

**Figure 8 bioengineering-11-00528-f008:**
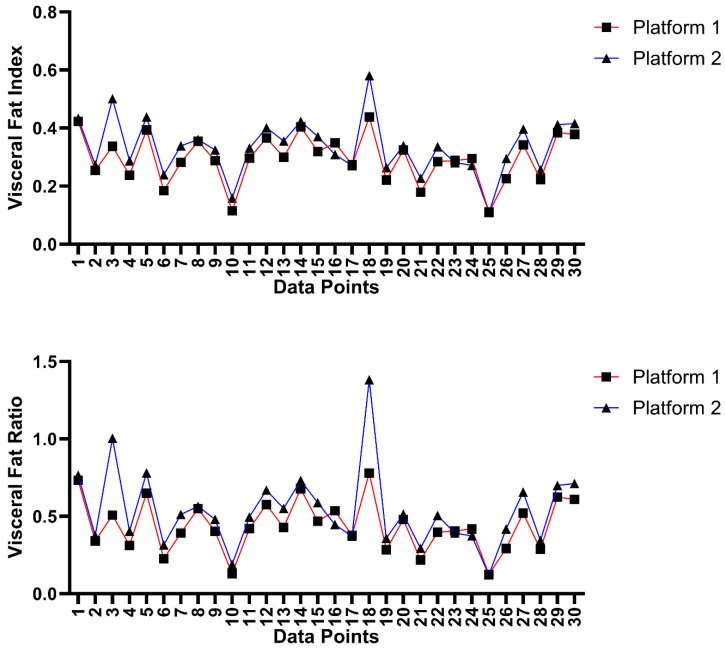
Inter-platform Analysis Graph depicting variation in visceral fat ratio and visceral fat index measurements between CoreSlicer (platform 1) and pMRI (platform 2). “Data Points” refers to scans taken at L5 and umbilicus levels for each of the 15 patients.

**Table 1 bioengineering-11-00528-t001:** Baseline characteristics of study patients.

Baseline Characteristics of Patients on Study (*n* = 32)	
**Age (years)**	
Mean ± SD	38.2 ± 14.1
**Sex**	
Female	18 (56.2%)
Male	14 (43.8%)
**Weight (kg)**	
Mean ± SD	73.2 ± 16
**Height (m)**	
Mean ± SD	1.72 ± 0.108
**BMI**	
Mean ± SD	24.8 ± 3.95

**Table 2 bioengineering-11-00528-t002:** Adipose tissue parameters from platform 1 at the level of umbilicus and L5. † Indicates non-normal distribution.

Parameter (*n* = 32)	Umbilicus	L5	*p*-Value
Subcutaneous Fat Area (SFA), mean (±SD)	239 ± 96.3 cm^2^	262 ± 97.3 cm^2^	0.349
Visceral Fat Area (VFA), mean (±SD)	106 ± 68.2 cm^2^	95.4 ± 45.8 cm^2^	0.8932 †
Visceral Fat Index (VFI), mean (±SD)	0.3 ± 0.105	0.273 ± 0.0881	0.256
Visceral Fat Ratio (VFR), mean (±SD)	0.465 ± 0.256	0.395 ± 0.177	0.2767 †

**Table 3 bioengineering-11-00528-t003:** Average fat volumes and ratio from platform 2 by reader and by level. *p*-Values shown are calculated overall between the two levels and are non-significant for all parameters. † Indicates non-normal distribution.

Parameter (*N* = 15)	Umbilicus Reader 1	L5 Reader 1	Umbilicus Reader 2	L5 Reader 2	*p*-Value between Umbilicus and L5 Levels
Subcutaneous Fat Volume, mean (±SD)	120 ± 54.4 cm^3^	131 ± 56.7 cm^3^	119 ± 52.7 cm^3^	127 ± 55.3 cm^3^	0.5
Visceral Fat Volume, mean (±SD)	61.7 ± 27.7 cm^3^	55.6 ± 19.5 cm^3^	71.3 ± 25.9 cm^3^	60.6 ± 17.2 cm^3^	0.2805 †
Total Fat Volume, mean (±SD)	182 ± 72	187 ± 64.6	191 ± 68.4	188 ± 64	0.95
Visceral Fat Ratio, mean (±SD)	0.562 ± 0.207	0.507 ± 0.289	0.683 ± 0.306	0.562 ± 0.282	0.06 †
Visceral Fat Index, mean (±SD)	0.349 ± 0.0869	0.318 ± 0.107	0.389 ± 0.101	0.344 ± 0.099	0.14

## Data Availability

The raw data supporting the conclusions of this article will be made available by the authors on request.

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
