# Peer review of "Quantification of Visceral Fat at the L5 Vertebral Body Level in Patients with Crohn’s Disease Using T2-Weighted MRI"

_bioengineering, 2024, doi:10.3390/bioengineering11060528_

Round 1
Reviewer 1 Report (Previous Reviewer 2)
Comments and Suggestions for Authors
The authors addressed all comments sufficiently.
Reviewer 2 Report (Previous Reviewer 1)
Comments and Suggestions for Authors
Dear Favour,
thank you for going into detail with the "criticisms" of the first version. I feel that the quality of the paper has been markedly improved now.
This manuscript is a resubmission of an earlier submission. The following is a list of the peer review reports and author responses from that submission.
Round 1
Reviewer 1 Report
Comments and Suggestions for Authors
In a rather small cohort of 32 included patients with Crohn’s disease, the authors quantify visceral adipose tissue (VAT) in two single (representative) slices at the umbilical- and L5-level acquired by a T2-weighted MRI technique. Visceral fat ratio (VFR, i.e. visceral fat divided by subcutaneous fat) and visceral fat index (VFI, i.e. visceral fat divided by total fat) are calculated, fat areas are quantified by using two image analysis platforms in a manual approach. Interrater- and inter-platform correlations are determined and, as a result, significant associations for VFI and VFR are shown with a good agreement between investigators and a good/moderate agreement between platforms. The authors conclude, that the L5 level can serve as an alternative anatomical landmark to the umbilical lever for quantification of VAT and that the applied T2-HASTE technique can be used for this purpose.
Major comments:
Although quantification of VAT is of increasing interest in obesity research, the approach and information/results of this paper are somewhat outdated (sorry for this diction) in my opinion, as (1) it is well known from literature that different positions in the body trunk can be used to – more or less – estimate the volume of VAT (e.g. doi:10.1038/s41598-023-37245-3, doi:10.3945/ajcn.115.111203, doi:10.1038/oby.2012.53, doi:10.1097/RLI.0b013e3181f10fe1 just to name a few), (2) there is no reference standard, which location might be more suited to predict total volume of VAT and no reference imaging sequence (e.g. widely used fat selective Dixon MRI) for comparison, (3) just a more or less manual procedure for segmentation – VAT segmentation can be reliably performed applying AI-based approaches, (4) as mentioned above: just a small cohort.
There are some further comments given by line number:
Abstract:
1. 17: Feasibility of MRI for VAT measurement has been proven since decades
2. 28: this is a rather low agreement when comparing two segmentation strategies, I would expect associations higher than 0.95, otherwise one of the platforms detects something completely different. Same for interrater results what makes the manual approach somewhat arbitrary…
Introduction
3. 51: MRI is widely used for this purpose and intensity values are well known (e.g. in Dixon techniques where fat selective images are being recorded, in T1-weighted MRI where fat appears significantly brighter than all other tissues – which is, by the way, not the case for T2-weighted images…)
4. 60: neither the umbilical, nor the L3-level are optimal for segmentation as both levels do not correctly reflect total VAT volume – distribution of VAT differs between men and women and between young and old subjects (see, e.g., doi:10.1126/sciadv.add0433). You might “soften” this statement and use “accepted” instead of “optimal”
5. 69f: this sentence might be omitted as you already mentioned T2 HASTE as an alternative and there is no reference sequence (e.g. Dixon) for a direct comparison of results.
Materials and Methods
6. 87f: BMI might be shifted to line 83 – this is a common anthropometric measure and it must not be explained how it is calculated.
7. 94: weighted (just a typo…) and: please give information on sequence parameters for the HASTE sequence
8. 104f: why just a subset? In a cohort of 32 patients it should be no problem to segment all available data!
9. 111f: it is just an area, not a volume as you segment single slices!
10. 118-127: same as line 111ff, please delete!
11. 128ff: this is a rather long description with (partly) doubled information in 2.3 and 2.4 – should be shortened substantially.
12. Figure 1: please use images of the same patients for B and C!
13. 148: please name the freeware/software here
14. 149: neither the green nor the red line are axial – did you reconstruct tilted slices from a 3D-volume?
15. 175f: in line 80f you mentioned that those patients were excluded! Thus, this process is not necessary and – by the way – inaccurate. 2.5 can be removed!
16. 202ff: excluded, included – please get concrete! In my opinion you should just do analyses in patients completely fitting in the maximum field of view of the imager (please give information on this and on the scanner used!)
Results
17. 214: you mention several exclusion criteria (prior to segmentation) and should, thus, not start again with this topic in the Results section!
18. 229: basic information of Table 1 is more or less given in the text, thus there is no necessity to repeat this in a Table.
19. 230: please be aware that superficial is not the same as subcutaneous! There are two compartments of subcutaneous adipose tissue: superficial and deep which can be separated. As you did not do this: subcutaneous instead of superficial!
20. 241ff: avoid this interpretation in the caption!
21. Figure 6: please shift x- and y-axis as the umbilical level is your standard reference point.
Discussion
22. 325: perhaps I missed this information: where is it stated that the umbilical level is the optimal one? If not yet done: give a reference for this statement! Finally, the optimum is and remains a volumetric assessment!
Comments on the Quality of English Languageok
Author Response
Please see the attachment. Thank you for your comments.

Reviewer 2 Report
Comments and Suggestions for Authors
This manuscript describes a study in which T2w MRI was used for quantifying visceral fat at L3 vs L5 vertebral body. The aim was to test whether there were any statistical differences between the results arising from L3 vs those arising from L5. A secondary aim was to assess the feasibility of using T2 HASTE sequence for the aforementioned task.
Major Comments:
The paper shows that T2 HASTE, which is commonly used in clinical protocols, can be used for quantifying fat and that a DIXON sequence is not essential, per se. This is an important conclusion that would be of interest to readers. However, this conclusion highly depends on the quality of the volumetric calculation, which is unclear as outlined below:
The authors describe a method for estimating the portion of the subcutaneous fat that is out of the field of view. It is unclear whether this linear dimension was used for extrapolating the final, total volume of subcutaneous fat. If the single linear measurement was indeed used in calculating the total subcutaneous fat, then a detailed explanation as to how this linear measurement was used need to be added. As it stands, the accuracy of that approach is questionable since the exact shape of the abdomen outside of the FoV is unknown (its elliptical shape would need to be accurately fitted using more than 1 parameter to get an accurate shape and volume estimate).
Minor Comments:
1. Figure 1: maintain a consistent colour scheme between pane B and C. Since purple and yellow are used in all other images, continue to use those two colours in Fig. 1 pane C as well (instead of red and blue).
2. Lines 116-123 are redundant in that paragraph and should be removed.
3. Line 124 contains an incomplete / grammatically incorrect sentence "due its focus"
4. Image quality/resolution of Figure 3 needs to be improved.
Author Response

(The authors gave the same response as above.)
